



# Short-term prediction of extreme sea-level at the Baltic Sea coast by Random Forests

Kai Bellinghausen[1], Birgit Hünicke[1], and Eduardo Zorita[1]

[1]Institute for Coastal System Analysis and Modelling, Helmholtz-Zentrum Hereon

**Correspondence:** Eduardo Zorita (eduardo.zorita@hereon.de)

**Abstract.**

We have designed a machine-learning method to predict the occurrence of extreme sea-level at the Baltic Sea coast with lead times of a few days. The method is based on a Random Forest Classifier and uses sea level pressure, surface wind, precipitation, and the prefilling state of the Baltic Sea as predictors for daily sea level above the 95% quantile at seven tide-gauge stations representative of the Baltic coast.

The method is purely data-driven and is trained with sea-level data from the Global Extreme Sea Level Analysis (GESLA) data set and from the meteorological reanalysis ERA5 of the European Centre for Mid-range Weather Forecasting. These records cover the period from 1960 to 2020 using one part of them to train the classifier and another part to estimate its out-of-sample prediction skill.

The method is able to satisfactorily predict the occurrence of sea-level extremes at lead times of up to 3 days and to identify the relevant predictor regions. The sensitivity, measured as the proportion of correctly predicted extremes is, depending on the stations, of the order of 70%. The proportion of false warnings, related to the specificity of the predictions, is typically as low as 10 to 20%. For lead times longer than 3 days, the predictive skill degrades; for 7 days, it is comparable to a random skill.

  The importance of each predictor depends on the location of the tide gauge. Usually, the most relevant predictors are sea level pressure, surface wind and prefilling. Extreme sea levels in the Northern Baltic are better predicted by surface pressure and the meridional surface wind component. By contrast, for those located in the south, the most relevant predictors are surface pressure and the zonal wind component. Precipitation was not a relevant predictor for any of the stations analysed.

  The Random Forest classifier is not required to have considerable complexity and the computing time to issue predictions is typically a few minutes. The method can therefore be used as a pre-warning system triggering the application of more sophisticated algorithms to estimate the height of the ensuing extreme sea level or as a warning to run larger ensembles with physically based numerical models.



## 1 Introduction

Storm surges are an extreme and short-lived increase in sea level mainly induced by extreme atmospheric conditions of wind
(e.g. storms) and low-pressure systems (Wolski and Wisniewski, 2021; Field et al., 2012; WMO, 2011; Weisse and von Storch,
2010; Harris, 1963). They are a major natural hazard for coastal societies as they not only impose severe damage to infras-
tructure at coastlines but are also deadly to humans. Hence, monitoring and forecasting systems for storm surges are important
to prevent societal damage and inform decision-makers. This study explores the capability of short-term predictions (lead
time of a few days) of storm surges in the Baltic Sea using a purely data-driven machine-learning approach. Technically, the
storm-surge problem is an air-sea interaction problem, where the atmosphere forces the water body, which in turn responds
with oscillations of the water level at various frequencies and amplitudes. While the atmosphere and its wind-field influence
the currents and wave dynamics of the sea, the currents in turn influence the wave dynamics which again alter the wind field
(Gönnert et al., 2001). Hence, the underlying processes of storm surges are highly non-linear which makes forecasting and
predicting them cumbersome.

Operational forecasting systems of storm surges rely on numerical dynamical ocean-atmosphere models (WMO, 2011;
Gönnert et al., 2001). In the Baltic Sea, a couple of regional models are in operation, like the BSHcmod from the Bundesamt
für Schifffahrt und Hydrographie (BSH), which is a hydrostatic circulation model. While those dynamical models generate
reasonable estimations for general water level elevations, they often underestimate extreme (storm surge) events (Muis et al.,
2016; Vousdoukas et al., 2016). This is explained by an insufficient grid-resolution (Muis et al., 2016), which leads to a
misrepresentation of e.g. wind fields (WMO, 2011) and extratropical cyclones (Rutgersson et al., 2021). Furthermore, the
effect of mesoscale weather systems is not represented in current storm surge models as there are no networks providing data
at these scales (WMO, 2011). Dynamical models need to incorporate the complex transfer of (kinetic) energy imposed by the
atmosphere on the ocean by linking them at the boundary layer through a moving and changing interface (Gönnert et al., 2001).
Usually, the data of meteorological fields are interpolated in time and fed to the surge model grid (von Storch, 2014), which
may lead to too smooth short variability of the atmospheric forcing, which in turn may leave extreme events underrepresented
in the simulations. According to Muis et al. (2016), the underestimation of extreme events can also be explained by the lack of
non-linear coupling of storm surge drivers in dynamical models.

Alternatively to dynamical models, forecasting systems can be based on data-driven algorithms. These algorithms are not based
on equations representing the physical dynamics of a process but try to extract insight about it in form of emerging patterns by
analyzing observational data sets of the forcing (atmospheric and/or oceanic) and of the response (storm surge). This makes
them computationally more efficient than dynamical models (Harris, 1962), at the expense of being a method oblivious of
the underlying physical mechanisms. Besides the classical statistical methods based on simplified statistical models of the
underlying processes, Machine Learning (ML) is one example of data-driven algorithms and is becoming more popular in
climate sciences. Machine Learning (ML) algorithms are usually more complicated than classical statistical methods and do
not attempt to explicitly represent physical processes but rather try to identify recurring patterns in the data that may be
used for predictions. Those complex and not obvious links between predictors and predictands, and this interesting property





contributes to their growing application. However, this very complexity makes them more difficult to interpret than classical methods. Also, special care is therefore needed to avoid statistical pitfalls, such as overfitting. Several studies applied ML-methods in order to analyze and predict storm surges with promising results (Tiggeloven et al., 2021; Bruneau et al., 2020;

Tadesse et al., 2020; Gönnert and Sossidi, 2011; Sztobryn, 2003). Statistical and machine learning models were compared when simulating daily maximum surges on a quasi-global scale based on either remotely sensed predictors or predictors obtained from reanalysis products like ERA5-Interim data Tadesse et al. (2020). The storm surge predictand was derived from two data sets, the observed hourly sea level data from the GESLA-2 database and other *in situ* data of daily maximum surges. They compared linear regression models to a machine learning method called Random Forest (RF)s. The authors found that

data-driven models work well in extratropical regions, e.g. the Baltic Sea, and that the ML methods generally performed better than linear regression. Storm surge prediction on a global scale has also been the focus of several ML-models Bruneau et al. (2020). They show that ML – in this case Artificial Neural Network (ANN)s – reconstructed storm surges with significant skill but still struggled to represent the strongest extreme events. Bruneau et al. (2020) explained this by unavoidable limitations of the training data, as extreme events are only a small fraction of the available data set. Because ANNs are trained with a

procedure that is ill-designed for outliers and biased towards the representation of the average dynamics, extreme surges can not be reliably reproduced. Tiggeloven et al. (2021) use a variety of deep learning methods, a subbranch of ML, to investigate storm surges at 736 tide stations globally. The overall result showed that ML approaches to capture the temporal evolution of surges and outperform a large-scale hydrodynamic model. However, extreme events were underestimated due to similar reasons as found by Bruneau et al. (2020).

Most approaches using ML-methods are global and hence lack specificity for the Baltic Sea basin. The only study (to our knowledge) that applied ANNs specifically to the Polish coast of the Baltic Sea was undertaken by Sztobryn (2003), using preceding mean sea level as well as wind speed and wind direction as predictors of high water levels. She showed that neural networks can be successfully integrated into operational forecast services and possibly reduces their average error. Similar to the global studies, the study by Sztobryn (2003) showed an underestimation of extreme water levels. Altogether,

a thorough application of ML to predict extreme storm surges at several tide-gauging stations in the Baltic Sea is missing in current literature. Hence, we will render a simple RF to the specific storm-surge drivers of the Baltic Sea in order to predict extreme storm surges defined by the top five percent highest measurements of sea-level taken from the Global Extreme Sea Level Analysis (GESLA)3-project (Haigh et al., 2021). The Baltic Sea is known for broad coverage in atmosphere and ocean measurements (Rutgersson et al., 2021) thus being a very good testbed for ML-models.

## 85 1.1 Specific characteristics of the Baltic Sea

Apart from the atmospheric forcing, the amplitude of storm surges also substantially varies with specific local conditions like the topography of the ocean basin, the extent of ice cover, as well as the direction of the storm track surpassing the basin and the shape and orientation of the coastline (Muis et al., 2016; WMO, 2011; Weisse and von Storch, 2010; Gönnert et al., 2001). Hence, it is necessary to understand the local characteristics of the Baltic Sea when building and interpreting a storm-surge

model.



The Baltic Sea is a semi-enclosed intracontinental sea of the Atlantic Ocean that ranges from around 10°E - 54°N to 29°E - 65°N in Northern Europe (Weisse and Hünicke, 2019) as depicted in Fig. 1. It is connected to the North Sea and thus the Atlantic via the Straits of Denmark and the Kattegat. This connection plays an important role in the context of storm surges and tides. The Straits of Denmark block tidal waves and allows mainly internal tides of only a few centimeter within the Baltic Sea

(Rutgersson et al., 2021; Wolski and Wisniewski, 2021). Due to the very narrow connection to the Atlantic, storm surges are only induced internally (Weisse and Hünicke, 2019). According to (Eakins and Sharman, 2010) the volume of the Baltic Sea is about 20,900 km$^3$, which is also influenced by the inflow of water through the Straits of Denmark. Depending on the filling of the Baltic Sea, its depth changes. The average depth is about 55m, due to extensive shallow coastal areas (Weisse and von Storch, 2010; Leppäranta and Myrberg, 2009). The risk of storm surges considerably depends on the location due to the large

meridional extent of the Baltic Sea and the different orientation of coastlines (Hünicke et al., 2015). Weisse (2014) showed that north-eastern subbasins like the Gulf of Riga, the Gulf of Bothnia, and the Gulf of Finland are most likely to experience storm surges (bay effect). This is explained by the shape of these subbasins in combination with the eastward trajectories of low-pressure systems and strong westerly winds (Rutgersson et al., 2021; Wolski and Wisniewski, 2020; Holfort et al., 2014). By contrast, the central parts of the Baltic and the Swedish coast do not undergo strong variations in extreme water levels

(Rutgersson et al., 2021; Wolski and Wisniewski, 2020). Variations in the southwestern water levels of the Baltic can lead to positive and negative storm surges. Seasonally, the strongest increase in water levels is expected from September to February. Those winter half-year surges are mainly driven by processes that alter the volume of the Baltic Sea, e.g. prefilling, and by the ones that redistribute internal water masses of the basin, e.g. effects of wind (Weisse and Hünicke, 2019; Weisse, 2014; Hünicke and Zorita, 2006; Chen and Omstedt, 2005).

Apart from general drivers of storm surges like wind stresses, atmospheric pressure, and the tides (Harris, 1963) there are additional processes specific to the Baltic Sea that may lead to storm surges. These processes are due to the Baltic Sea being a semi-enclosed basin and range from the current state of water volume (prefilling) over the influence of climate modes like the North-Atlantic-Oscillation (NAO) to the extent of ice-sheet coverage (Wolski and Wisniewski, 2021). Hence, we summarize here the most important general drivers (wind stresses and atmospheric pressure) as well as specific drivers like the prefilling

and precipitation in this study.

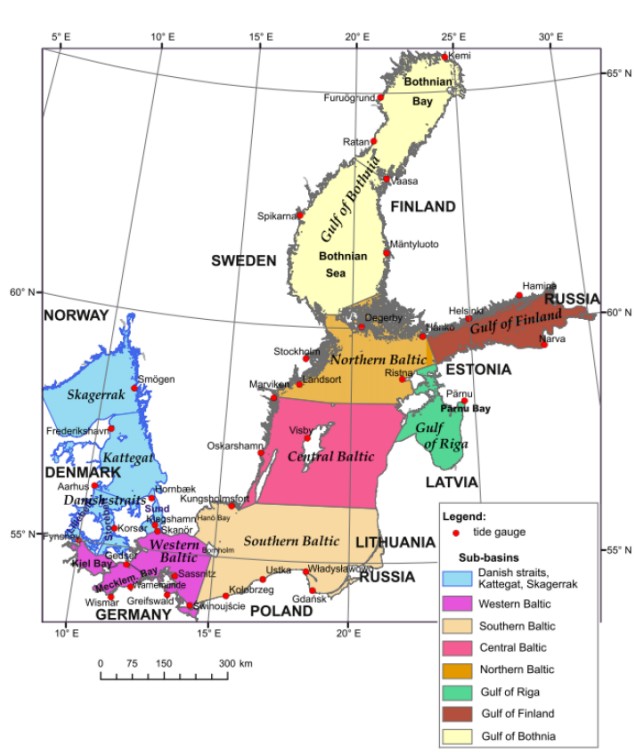

**Figure 1.** Subbasins of the Baltic Sea as indicated in (Wolski and Wisniewski, 2020).



Storm surges generated by the impact of wind stress are called *wind-driven storm surges*. The wind stress within a given area is further specified by the wind direction, its velocity, duration and fetch (Weisse, 2014). If a wind blows consistently over several days, it deforms the sea surface and causes drift currents and wind setup which eventually lead to a storm surge (Wolski and Wisniewski, 2021; Harris, 1963). Wind conditions in the Baltic Sea are mainly governed by the Westerlies and

the cyclonic activity in the Northern Europe-Baltic Sea area. Both of them are strongly influenced by the NAO-index, which explains parts of the variability of wind conditions in the Baltic (Donat et al., 2010). This is true especially during the winter months, where the winds are blowing (on average) from south-western directions (Weisse, 2014; Leppäranta and Myrberg, 2009). When strong westerlies stop blowing, the elevated sea surface in the northeastern parts of the Baltic Sea relaxes and water masses flush back towards the southern and southwestern coasts. This seiche-like fluctuation may raise the water levels

in the corresponding coasts (Weisse and von Storch, 2010). Furthermore, south-westerly winds, if maintained for several days, can cause a strong inflow of water masses into the Baltic Sea via the Straits of Denmark, leading to a condition of prefilling (Gönnert et al., 2001). Hence, the wind direction is an important indicator for the onset of storm surges at specified coastlines (Andrée et al., 2022; Wolski and Wisniewski, 2021).

### 1.1.1 Atmospheric pressure

Variations in atmospheric pressure are also a factor for storm surges. The Baltic Sea is situated within the location of the *circumpolar low-pressure zone* (Gönnert et al., 2001). Low-pressure systems are mostly associated with regions of less than 980 hPa (Wolski and Wisniewski, 2021; Holfort et al., 2014). If such a system moves at relatively high velocities (greater than 16 ms$^{-1}$) a *subpressure-driven storm surge* occurs (Wolski and Wisniewski, 2021). This is due to the *inverted barometer effect* which causes a rise of the sea surface underneath low-pressure systems (Weisse and von Storch, 2010), eventually inducing

a *baric wave* traveling along the trajectory of the system (Wolski and Wisniewski, 2021). In hydrostatic equilibrium, a drop in pressure of 1 hPa increases the sea level by about 1 cm (Wolski and Wisniewski, 2021; Harris, 1963). As low-pressure systems in the Baltic Area usually move from the (South-)West towards the (North-)East, the water surface is more frequently elevated in the North and depressed in the South (Wolski and Wisniewski, 2021). This is why a sea level decrease at Western Baltic tide gauging stations could be seen as an indicator for storm surges generated by low-pressure systems. Nevertheless,

the pressure effect should not be considered to be isolated from the wind effect. As Wolski and Wisniewski (2021) state, there is synchronous activity of both, pressure and wind, during every storm surge and they label these situations as *mixed surges* or *subpressure-wind surges*. On the one hand, these two forces combined may amplify the storm surge and increase its intensity, on the other hand, they may cancel each other out and decrease the severity of the storm surge (Wolski and Wisniewski, 2021).

The changing total volume of the Baltic Sea is also important for storm surges. The Baltic Sea contains an averaged volume

of 20.900 km$^3$ (Eakins and Sharman, 2010) that is constantly altered due to different in and outflows (Weisse and Hünicke, 2019). The main inflow is the saltwater exchange of the North Sea and the Baltic Sea via the Straits of Denmark, which is approximately 1180 km$^3$a$^{-1}$ (Leppäranta and Myrberg, 2009). On a daily basis, up to 45 km$^3$ are exchanged between the basins in both directions. Evenly distributing this water mass over the whole Baltic Sea would correspond to a sea level change of 12 cmd$^{-1}$ (Mohrholz, 2018)) or 320 cm averaged over a whole year (Leppäranta and Myrberg, 2009). If the water level





of the Baltic Sea is elevated 15 cm above the mean sea level for more than twenty consecutive days due to increased inflow via the Straits of Denmark, Mudersbach and Jensen (2010) speak of a *prefilling* or *preconditioning* of the Baltic Sea. The degree of filling is then given by the averaged water level of the Baltic Sea (Weisse, 2014). Usually, the tide gauging stations in Landsort (Sweden) or Degerby (Finland) are used as proxies for measuring prefilling (Weisse, 2014; Janssen et al., 2001). Depending on the degree of prefilling, storm surges can become more likely and extreme as less wind is needed to induce

wind setup (Weisse and Weidemann, 2017; Weisse, 2014). It is mainly the already mentioned south-westerly wind direction that, when blowing over extended periods, leads to an increased inflow of water masses to the Baltic Sea through the Kattegat (Wolski and Wisniewski, 2021; Hünicke et al., 2015; Weisse, 2014). As the inflow is predominantly depending on the strength of the Westerlies, which themselves are subject to the NAO-Index, years of a strong positive NAO-Index tend to be associated with higher numbers of prefilling in the Baltic Sea (Weisse, 2014). On shorter time scales, the water exchange is mainly

driven by atmospheric conditions (Leppäranta and Myrberg, 2009). For example, a sequence of fast-moving low-pressure systems coming from the West and travelling to the North-East of the Baltic Sea resulted in strengthened inflows (Wisniewski and Wolski, 2011). According to Leppäranta and Myrberg (2009) peak months of inflow are during winter, especially from November to January, with a monthly inflow of 120 km$^3$. Combined with the effects of stronger winds and rainfall in winter, the preconditioning is an important driver of storm surges.

Finally, when low-pressure systems and corresponding cyclones move over the Baltic Sea, they usually bring precipitation along (Leppäranta and Myrberg, 2009; Harris, 1963). Extreme precipitations associated to low-pressure systems are most frequent in winter (Rutgersson et al., 2021). The largest amount of precipitation is found at the eastern coast of the Baltic Sea due to the winds blowing mostly eastward in winter time (Leppäranta and Myrberg, 2009). As stated by Weisse and Hünicke (2019), heavy precipitation increases the total volume of the Baltic Sea and changes the density due to a change in

salinity profiles, which combined may lead to an increased overall water level. Therefore the influence of precipitation is not directly related to storm surge magnitudes, but rather alters preconditions like the prefilling of the Baltic Sea and the filling of rivers and estuaries (Gönnert et al., 2001). As Harris (1963) state, precipitation can lead to above-normal water levels in estuaries which alters the gradient in the river water level and thus leads to an accumulation of rainwater in the river bed. Additionally, if the river bed is relatively flat, the storm surge can easily penetrate the riverbanks upstream for tens or even

hundreds of kilometers (WMO, 2011). In the worst case, extreme surges can propagate over marshes and expand them. Winds blowing over these open waterbodies generate waves and hence transport even more water inland. Hence, indirect effects of precipitation combined with the onset of a storm surge can lead to severe compound floodings in the Baltic Sea, especially in low-lying coastal areas (Rutgersson et al., 2021; Bevacqua et al., 2019).

In Section 2 we will further specify the underlying datasets of this study as well as their preprocessing. In Section 3 the

model architecture is presented and the basic principles of a RF are discussed. Furthermore, we will specify how the model was tuned and evaluated. In Section 4 we describe all conducted experiments and their reasoning while Section 5 summarizes their results. We end the study with a discussion and conclusion.


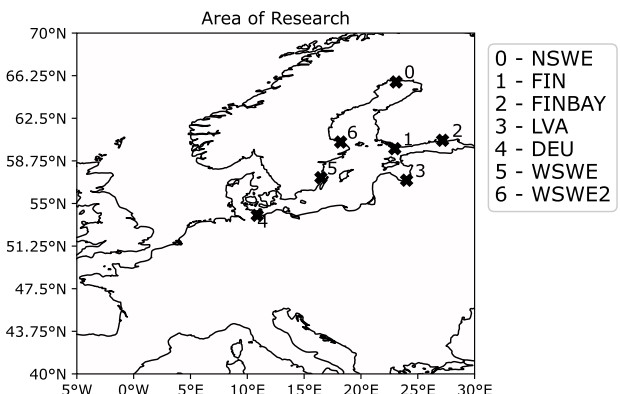

**Figure 2.** Map of whole research area. Crosses and numbers indicate the analysed stations within the Baltic Sea.

.

## 2 Data

The Baltic Sea provides one of the densest tide-gauge networks with records starting in the 19th century (Hünicke et al., 2015),
which is part of the record compilation of the Global Extreme Sea Level Analysis (GESLA) dataset. Together with the vast
European Re-Analysis (ERA5) dataset from Hersbach et al. (2018) it makes the sea a perfect test-bed for machine learning.

### 2.1 Area of research

The investigated area ranges from 5°W to 30°E and 40°N to 70°N as depicted in Fig. 2 and includes the Baltic Sea (BS). More
specifically seven stations were selected for model analysis. These stations are part of the GESLA data set. Station codes are
provided in Table A1. This set of stations should be representative of the coastal orientations and bays of the Baltic Sea.





## 2.2 Predictand

The GESLA dataset provides a global set of high frequency (at least hourly) sea level data with integrated quality control flags (Haigh et al., 2021). Height units of all stations were converted to metres and the time zone was adjusted to Coordinated Universal Time (UTC). A more thorough description of the compilation can also be found in Woodworth et al. (2016) and 
Haigh et al. (2021). The data is publicly accessible at http://www.gesla.org.

All stations we selected for model analysis contain hourly data, covering the period from 1960 to 2020. This sea level data is later used to derive the occurrence of storm surges at respective stations after preprocessing (see Section 2.4).

## 2.3 Predictors

Extreme surges are induced by low-pressure systems, precipitation, wind fields and prefilling of the BS. Except prefilling, we 
incorporated each of those drivers as predictors into our model setup based on atmospheric European Re-Analysis (ERA5) data provided by the European Centre for Medium-Range Weather Forecasts (ECMWF). The reanalysis combines model and observational data leading to a dataset with a high temporal and spatial resolution. The ERA5 dataset ranges from 1959 to present with hourly estimates of atmospheric variables and is spatially resolved on a 30km (approximately 0.27 degrees) grid covering the Earth (Guillory, 2017). We select the period from 1999 to 2020 for this study. All variables of ERA5 used as 
predictors are shown and briefly described in Table A2. They are surface pressure (SP), total precipitation (TP), eastward wind at 10m height (U10), northward wind at 10m height (V10). Each variable is extracted from the two-dimensional field depicted in Fig. 2. Additionally, we implemented a predictor of prefilling by using the GESLA timeseries of sea-level data at the station of Degerby as a proxy (Weisse, 2014; Janssen et al., 2001).

## 2.4 Preprocessing

We deduce the predictand from the GESLA dataset, as follows. We only incorporated data labeled by the GESLA project as *analysis data*. This data is either not quality controlled (control flag 0), has correct values (control flag 1) or is an interpolated value (control flag 2). We used the interval 1999 to 2008 to train the algorithm and reserved the period 2009 to 2018 for testing. Only for stations 3 and 4 the period from 2009 to 2018 was chosen as training data because the records originate from 2005 in the GESLA dataset. In order to obtain a *stationary process* we temporally detrended both timeseries, for training and testing 
period separately, via a linear least-squares regression. We then selected seasonal data of the winter months December, January and February. We did so because the strongest increase in water levels is expected from September to February. From this seasonal selection, we then apply *one-hot-encoding*, i.e. data points with values above the 95th-percentile of the timeseries are converted to 1 (i.e. extreme storm surge), all others are set to 0 (i.e. no extreme storm surge). This implies our definition of extreme storm surges being the top 5% highest hourly recorded water levels at one particular station. Finally we converted the 
hourly recording frequency to daily time-frequency by taking the maximum value of the timeseries per day. Hence only one incidence of an extreme storm surge during the day marks the whole day as an extreme event.



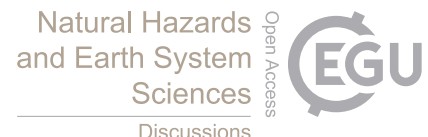

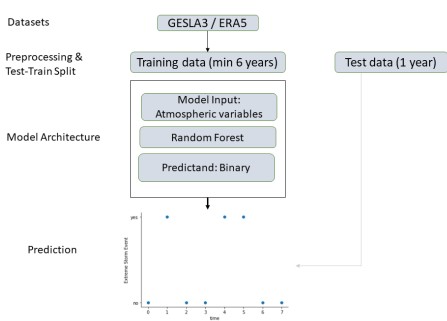

**Figure 3.** Software architecture as a blueprint based on: Tadesse et al. (2020)

.

According to Bruneau et al. (2020) six years of daily input data is sufficient for ML algorithms to produce reliable predictions. Hence, for each predictor, the years 1999 to 2008 were selected to train the model. We used *Climate Data Operators*
*(CDO)* to extract predictor values within the area of research depicted in Fig 2 and to further calculate daily averages of the hourly recorded data.

The only predictor that is not based on the ERA5 dataset and hence was treated differently during preprocessing is the prefilling (PF) of the Baltic Sea. This predictor is extracted from the GESLA dataset at the station of Degerby. If PF is combined with ERA5 predictors, the time-frequency of ERA5 is used. Hence, we reduced the hourly data of the station to daily data by using
the maximum recorded water level of that day as an entry. If PF is used as a single predictor, the frequency is kept as hourly. The timeseries of the predictor and predictand are intersected by date and a timelag is introduced to the predictor by shifting its timeseries accordingly. All Not-a-Number (NaN) entries were set to -999, which ensures that the ML algorithm treats those entries as outliers and ultimately neglects them. Note, that the predictor is a spatial map and possibly needs a further reduction in dimension depending on the model infrastructure at usage before starting the model training.

## 3 Methods

The overall structure of our software is sketched in Fig. 3. After preprocessing the datasets, they are separated into training (75%) and test data (25%) used to fit and tune the model. We fit the model with similar combinations of predictors for each station. The model then processes the atmospheric predictors (denoted features in ML parlance) in order to provide a binary prediction about the extreme storm surges (predictands, also called labels). Our software is publicly accessible on GITHUB
(Bellinghausen, 2022) and is based on the *scikit-learn* library of Python.



## 3.1 Random forests

As a classifier, we used the *RandomForestClassifier* from *scikit-learn*. A thorough description of RFs can be found in Müller (2017) and Géron (2017), from which we will briefly discuss the most important points.

The model architecture of a RF is based on an ensemble of Decision Tree (DT)s. DTs rely on a hierarchy of if/else-questions in
order to conclude with a prediction. A simplified example is shown in Fig. 5. In this case, the DT formulates sequential if/else-questions about the predictors U10, SP and PF. The grey nodes indicate a path of input data, where each question is answered positively, hence leading to the prediction of an extreme storm surge. In reality, the questions in each node are more complex, testing for continuous values of the predictor at hand (e.g. $u10 > 17ms^{-1}$ as a test for strong westwind at a specific point within the research area). The rectangular node is called *terminal node* or *leaf* and provides the prediction. When fitting data
to a decision tree, the algorithm essentially finds the best sequence of if/else- questions to get to the true answer. A prediction on unknown predictor data is then made by sifting through the optimized DT, answering all if/else questions.

In a RF (Fig. 4) as an ensemble of Decision Tree (DT)s, each of those DTs processes a random sample of the test data in order to conclude with a prediction. In general, those predictions are then averaged in order to get the overall prediction of the RF. For a binary classification problem, as in our case, this averaging is done via a majority voting.

One can get valuable insight on which predictors the DT relied the most via the concept of *feature importance*. This concept assigns a value between 0 and 1 to each feature, with higher numbers indicating greater importance. The importance of one predictor is estimated by computing the predictive loss of the algorithm when that predictor is omitted. The value of the importance is normalized by requiring that the sum of all feature importances within a DT is 1. We will use this concept in order to investigate which regions within the research area are of importance for individual predictors (see Section 3.3).

One problem of DTs when using a large number of predictors and a small? the training sample size is that they are prone to *overfitting*, i.e. they may focus on representing the noise in the data representing the training data almost perfectly while performing much worse on test data. The risk of overfitting can be diminished by using RFs, which average the results of multiple independent and randomly built DTs. Using more DTs when building a RF leads to less overfitting.

The number of DTs used within a RF and the number of features analyzed when looking for the best split are just two examples
of so called *hyperparameters* that can be tuned when building a RF.





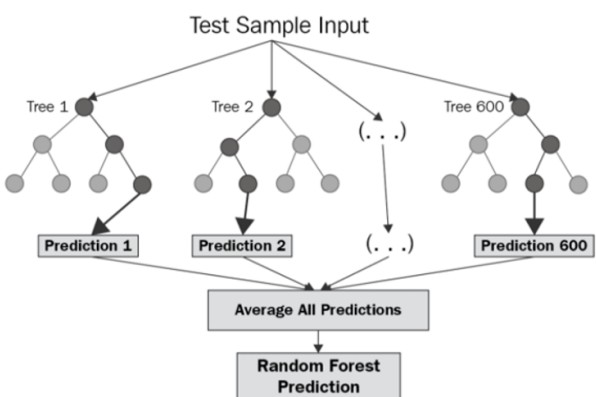

**Figure 4.** Model architecture of a Random Forest (RF) taken from https://levelup.gitconnected.com
.





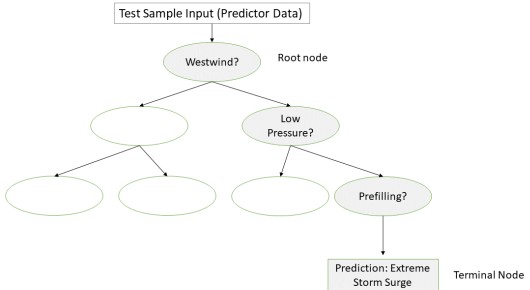

**Figure 5.** Simplified model architecture of a Decision Tree (DT). Grey Nodes indicate the path of test-data while sifting through the DT. Right pointing arrows refer to positive answers.




## 3.2 Model tuning

The *RandomForestClassifier* can be tuned in several ways by altering its hyperparameter (HP)s. For a RF the most important HPs control the amount of DTs used (*n_estimator*), the maximum depth of each DT (*max_depth*) and the number of features used when calculating the best split (*max_features*). In general, a larger value for *n_estimator* will lead to less overfitting as
the results of many DTs are averaged. With increasing *max_depth* the DTs get more complex, hence overfitting is more likely. The *max_features* controls the randomness of each DT with a smaller value reducing overfitting (Müller, 2017). While we set *max_features* to its default value of $\sqrt{n_{\text{features}}}$, we varied the other two.

In addition to those HPs we altered the following: *class_weight*, *oob_score* and *random_state*. The *class_weight* is used to associate weights with classes. This is particularly important in this study as we deal with extreme storm surges. Hence the
predictand dataset is unbalanced as there are by definition 95% of data points of class 0 (no extreme storm surge) and only 5% of class 1 (extreme storm surge). Setting the *class_weight* to "balanced" adjusts weights inversely proportional to class frequencies in the input data, i.e. the model will penalize more heavily wrong predictions about extreme storm surges than wrong predictions about normal conditions. The *oob_score* is set to "True", meaning that *out-of-bag*-samples are used to calculate scores for each DT within the RF. We also set the *random_state* to 0, which gives us and the reader the possibility to
reproduce results. All other HPs were left at their default value.

For the HPs *n_estimator* and *max_depth* we started out to manually guess their value based on the concept of *validation curves*. As this method got too cumbersome we switched to automatically finding the best combination of HPs using *GridSearchCV* and *RandomSearchCV*, two optimization procedures within *scikit-learn*. One can pass a list of values for each HP to these functionalities and they automatically search for the best combination using cross-validation. While *GridSearchCV* searches
among all possible combinations of HPs and their value lists, *RandomSearchCV* subsamples these lists and only uses a fixed number of parameter settings. This optimization is computationally expensive, especially when using *GridSearchCV*. Hence, we decided to only use *RandomSearchCV* in order to make multiple forecasts for all stations.

After several attempts we realized that many predictions were prone to overfitting, hence we strongly reduced the values of the *max_depth* parameter, drawing only from the list $[1, 2, 3]$. For the *n_estimator* we used either 333, 666, or 1000.
All settings are summarized in Table A3 for replication purposes.

## 3.3 Model evaluation

A common tool to evaluate binary classification models is the Confusion Matrix (CFM) (see Fig. 6). It summarizes the accuracy of a model in terms of rates. For our study, we aim for a high *True Positive Rate (TPR)*, which relates the absolute number of correctly predicted extreme storm surges $n_1$ to all incidences of storm surges $n_\epsilon$ in the underlying data. In Fig. 6 for
example $n_1 = 29$ out of $n_\epsilon = 40$ extreme storm surges were correctly predicted, leading to a TPR of $\frac{n_1}{n_\epsilon} = 72.50\%$. A high TPR automatically leads to a low False Negative Rate (FNR) since their sum equals one. The FNR indicates how often the model actually fails to predict a storm surge. With a high FNR the model can not be trusted as it very likely produces false predictions of security. Especially for extreme events this can lead to devastating damage to societies when protection measures





rely on model predictions with a high FNR, as eventually no measures are taken due to a model prediction of "no storm surge"

but in reality, an extreme surge appears.

The CFM can be evaluated on training and test data as well as on the validation set. If model predictions are correct almost always on training data, i.e. a TPR and True Negative Rate of around 100%, the model tends to overfitting. In practice, the CFM of test-data and the validation set is more interesting as it shows the performance of a model to unknown data. While for our study mostly the test-data set contains fewer cases of extreme storm surges than the validation set, the latter is more

interesting to look at. Note that for stations 3 and 4, no validation sets were used.

The second tool we use is a combination of the Feature Importance (FI) and a Predictor Map (PM). For each model, the importance of each feature is displayed by weights between 0 and 1 with all weights summing up to 1. Using FI lets us compare the overall importance amongst predictors when a combination of predictors is passed to the model. Furthermore, we can deduce which specific regions within the research area are important for model decisions for each predictor.

Unlike a correlation coefficient, the FI does not encode the magnitude of the feature that is indicative of the storm surge(Müller, 2017), e.g. whether it is low or high pressure within the area of importance that is related to a storm surge. Hence, we also include in the results the value of the predictor averaged over the cases that lead to a storm surge, denoted Predictor Map (PM). Only the top 1% area of importance regions is shown (grey squares), which was calculated by the 99th percentile of FI for each predictor. We investigated mainly two cases of PMs, namely True Positive Prediction (TPP) and False Negative

Prediction (FNP), which were compared amongst each other as in Fig. 7 (a) and (b). For instance, when PMs for TPP cases show low-pressure systems in the importance region while the FNP PMs only display high-pressure systems, this suggests that the model heavily relies on low-pressure systems to forecast a storm surge. By contrast, it also suggests that in some cases storm surges are induced even though there are high-pressure systems in the area of high FI.

To investigate further we calculated mean PMs for each of those cases and looked at the average difference of both maps as

depicted in Fig. 7 (c). As it is sufficient to only show maps for TPPs and the difference to FNPs we will do so in the results section.


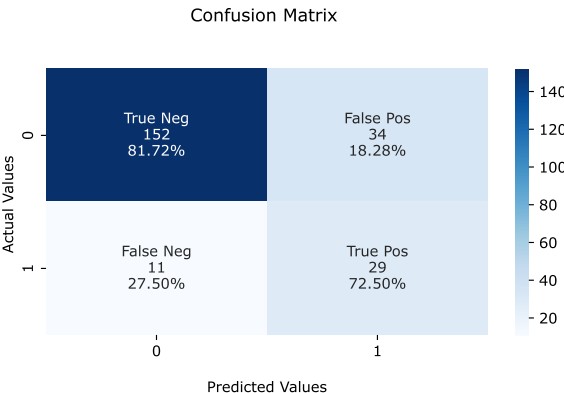

**Figure 6.** Confusion matrix for a binary classification model with absolute and relative values. The colour bar shows the maximum count of instances for all cases.

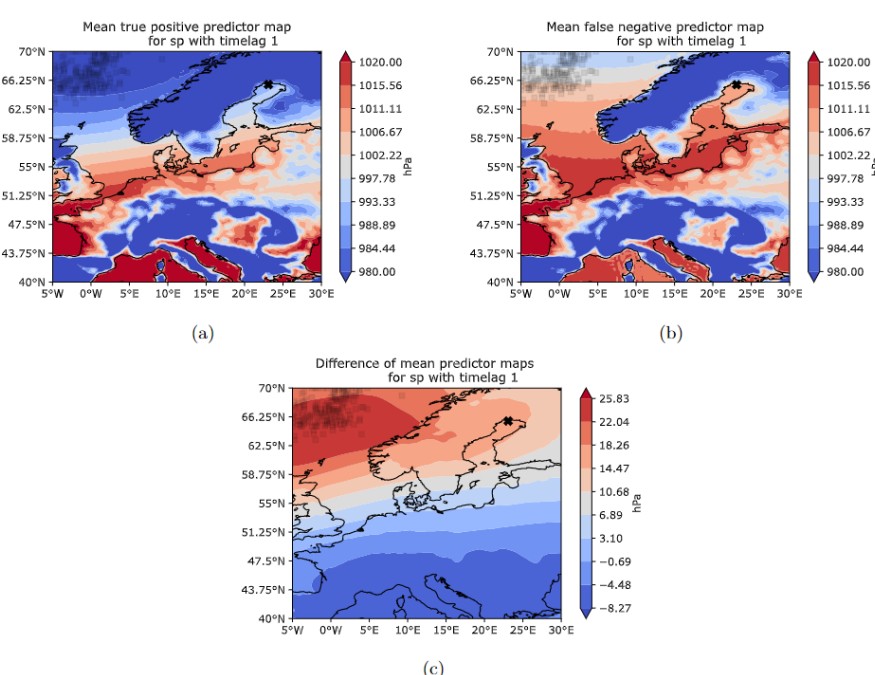

**Figure 7.** Mean Predictor Maps of SP all with timelag $\hat{t} = 1$ for station 0 (NSWE). (a) mean True Positive Prediction (TPP)s, (b) mean False Negative Prediction (FNP)s and (c) difference of both means FNP - TPP. Note the different scaling of the colour bar for the difference maps.



## 4 Model configurations

We build 6 overarching model configurations (**A – F**). For each configuration, we undertake subsets of model runs which are denoted by *run_ids*. All model runs are applied to each station, i.e. similar predictors and initial HP lists were used when
building the model for each station (note though, that the fitted model can be different for each station due to the automatic optimization of hyperparameters). As a starting point we analyzed the predictive skill of each predictor individually with time lags up to a week (**A**). Among those, time lags up to three days showed promising results. These time lags are interesting in order to predict storm surges in advance. Hence, we analyzed a combination of all predictors (ERA5 and PF) with time lags of 1 and 2 days, eventually getting insight on which predictors are most important (**B**). In experiment **C** we investigate the
coupling of strong winds and moving low-pressure systems by combining SP and U10 with various time lags. Because the west wind is an important driver of storm surges in the Baltic Sea due to the connection with the North Sea via the Kattegat and the possible wave build-up in the north-eastern region, we combined multiple time lags of U10 in experiment **D**. In **E** we looked into cumulative rain (TP with several time lags and U10), wind-induced waves in combination with prefilling and the state of prefilling induced by wind (both using U10 and PF). Since we use the water-level records at the Degerby station as a
proxy for prefilling and not the rolling mean of 20 consecutive days like Mudersbach and Jensen (2010), we combined several time lags (up to 30 days) of PF in experiment **F**.

All model runs are summarized in Tables A4 – A9.

## 5 Results

We selected promising results based on a combination of the TPR of the test and validation datasets, labeled as TTPR and
VTPR, respectively. When interpreting those rates, we will indicate the total amount of extreme storm surges for the specific dataset with $n_\epsilon$ in parenthesis. In contrast to ERA5-predictors, the prefilling dataset contains more instances of storm surges due to the hourly recording time, hence looking at TTPR instead of VTPR is also sensible.

The prediction skills described in the following subsections should be compared to simple *uninformed* prediction schemes. One such scheme could be to always predict storm surges. This scheme would display a true positive rate of 100% but also a false
positive rate of 95%. It would be very sensitive but very unspecific. A prediction scheme that always predicts 'no storm surge' would display true positive and false positive rates of 0% each. It would be totally insensitive. Both schemes are obviously not useful. A slightly more sophisticated scheme would issue a storm surge prediction randomly in 5% of the occasions. The true positive rate would amount to 5% and the false positive rate to 5%. It would be still rather insensitive. A prediction scheme has to clearly improve this random sensitivity and specificity.

### 5.1 A - Single Predictors and Multiple Timelags

We aim to find the best predictors for each station and analyze what timelags are most useful. Furthermore, we want to investigate how physical patterns of predictors change depending on the station location.





In terms of VTPRs, the SP leads to good results for all stations except DEU. For instance VTPRs were at 70.67% ($n_\epsilon = 75$) and 78.9% ($n_\epsilon = 109$) for time lags of one and zero days for stations NSWE and FINBAY, respectively. Only for station DEU the SP was not such a useful predictor (VTPRs below 50%). We could observe that for almost all stations time lags up to two days worked reasonably well, while longer time lags generally reduced the VTPR. The AoI changes depending on the station location. For instance, for WSWE and WSWE2 the AoI is in the Gulfs of Finland and Riga, while for NSWE and FINBAY it is in the European North Sea. Independent of the station, low-pressure systems are important within the AoI but for cases of FNPs the pressure rises several hPa. At station NSWE for example the AoI of SP is within the region 5°W – 5°E and 62.5°N – 70°N (see Fig. 8). Here, mainly low-pressure systems with less than 980 hPa lead to a correct prediction of a surge. The model tends to False Negative Prediction (FNP)s once the pressure in the AoI increases by a mean of around 25 hPa. In some cases, high pressure systems of more than 1020 hPa occurred in the AoI for FNPs. This behavior repeats for several stations.

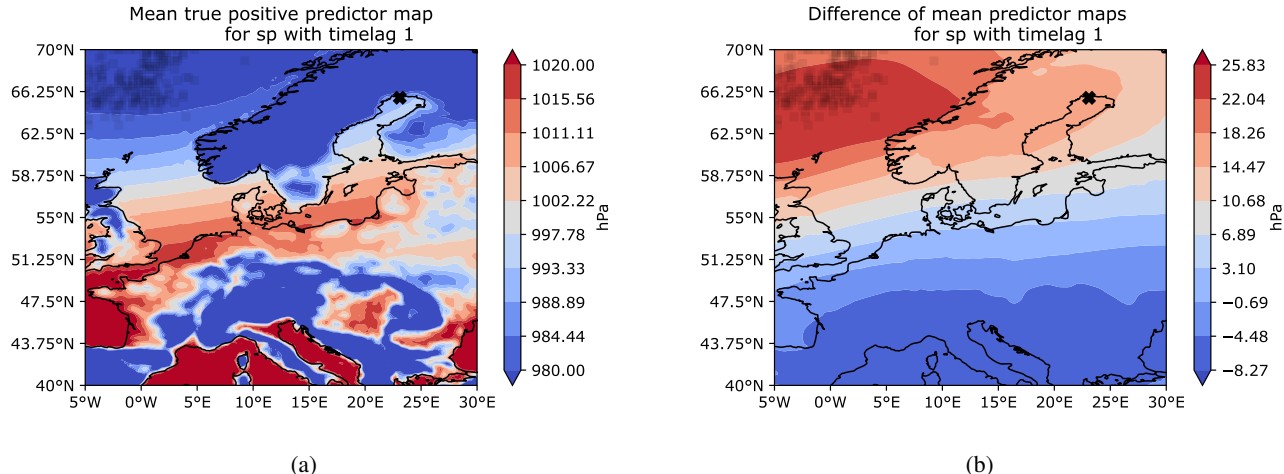

**Figure 8.** Mean Predictor Maps of SP with $\hat{t} = 1$ for station 0 (NSWE) for (a) TPPs and (b) the difference of FNPs and TPPs. Note the different scaling of the colour bar for the difference maps.





Model results show that west wind U10 is mostly useful for stations located in the eastern parts of the Baltic Sea. While the VTPRs for station NSWE are greater than 60%, VTPRs of 73.39%, 72.48% and 69.62% ($n_\epsilon = 109$) were measured for station

FINBAY. Those results were deduced for timelags zero, one and two respectively, indicating the short-term wind-fields as most relevant predictors. The AoI strongly depends on the location of the station as well as on the chosen time lag. For instance, for station FINBAY the AoI clusters around the Gulf of Riga and only covers the Region of the Kattegat lightly when using no time lag. This is interesting, as one would expect important short-term wind fields close to the station itself or at least close to the entrance of the Gulf of Finland, s.t. wind setup can be induced. Conversely, the AoI for time lags of one and two days

locates more around the Kattegat area and the Southern Coast of the Baltic Sea (see Fig. 9a – 9c). When looking at the AoI of station NSWE for a time lag of one day, it is not anymore the area around the Kattegat that is of importance but rather the west winds close to the UK over the North Sea (Fig. 9d).





**Figure 9.** Mean predictor maps for TPPs using U10 with time lags 0, 1, 2 for stations FINBAY (a – c), NSWE (d) and DEU (e, f).





Regardless of the AoIs location, the main wind direction is eastward. For instance at station FINBAY mean west wind speed of around $12\text{ms}^{-1}$ occurs in parts of the AoI for TPPs, especially around the Danish Straits. When looking at PMs separately,
windspeeds of $17\text{ms}^{-1}$ and more (i.e. storms) could be detected. Comparing the maps of TPPs to the ones of FNPs one can see that the model generally leads to false predictions when west wind fields become weaker. The difference maps show a mean decrease of west wind speeds of around $7\text{ms}^{-1}$ in parts of the AoI. Hence, the model is not as reliable when west winds are not strong, no winds or even east winds occur. This behavior repeats for other stations except for station DEU. Here, the east wind is used for model predictions (see Fig. 9e), which is an expected result. There are fewer (positive) storm surges at the German
Coast of the BS compared to other Bays as usually southwesterly winds lower the water level in those regions (Weisse and von Storch, 2010). It is interesting to see though that important short-term winds are mostly westward close to the station. This is also theoretically explained by the induced pile-up effect of wind at this station. Another mentioned driver for storm surges along the German Baltic Coast are seiches. These might be induced by the pronounced west wind around the station and over the Baltic Sea for a time lag of seven days. The long time lag could be sufficient for a wave growth towards the opposing coast,
which in turn leads to seiches once the wind turns westward or stops blowing (Fig. 9f).

While the zonal wind is a good predictor for stations located at zonal boundaries of the Baltic Sea, the meridional wind component V10 is a good predictor for stations located at the northern extent of the Baltic Sea. For instance, at station NSWE the meridional wind with a time lag of one day leads to VTPRs of 73.33% ($n_\epsilon = 75$). For all other stations meridional wind was not a good predictor. For instance, WSWE and WSWE2 only had VTPRs of around 50% using V10. At NSWE though
mainly light southwinds of around 6 ms$^{-1}$ are used by the model for True Positive Prediction (TPP)s, while it struggles when no meridional wind is blowing in the AoI.

Total precipitation was (in most cases) not a promising predictor. Except for station FINBAY, where it resulted in VTPRs of 69.72% and 72.48% (both $n_\epsilon = 109$), for no time lag or a time lag of one day respectively. The AoI for TP without a time lag is close to the station itself. When increasing the time lag by only one day, the AoI shifts towards the area around Bergen,
sometimes showing connecting patterns of importance across the North Sea towards the United Kingdoms (see Fig. 10). This behavior repeated for other stations but VTPRs were lower. Nevertheless, throughout all experiments, TP was not showing any consistent patterns in terms of PMs.




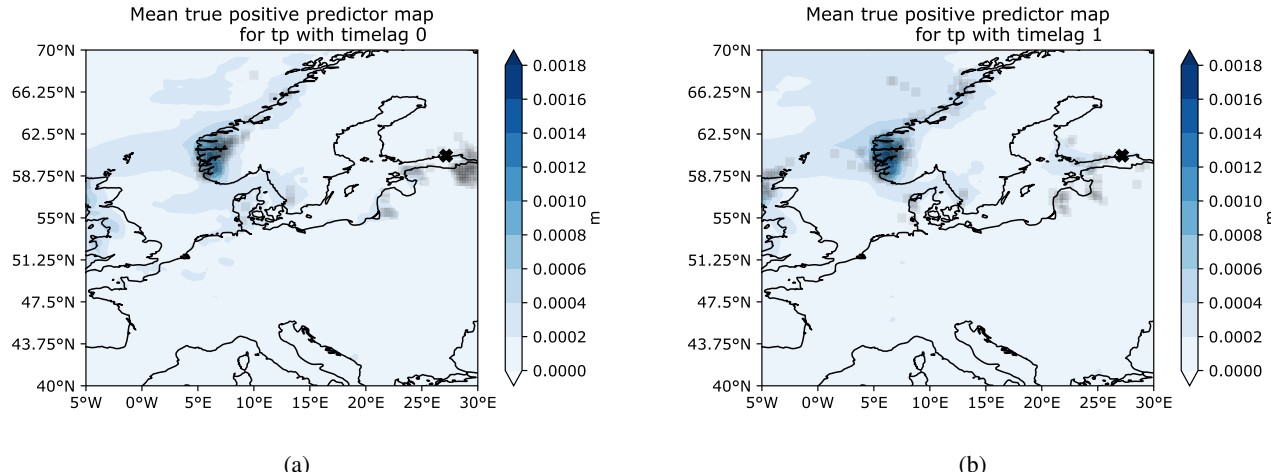

(a)

(b)

**Figure 10.** Mean Predictor Maps of TPPs for predictor TP with time lags 0, 1.





Despite the good performance of ERA5-predictors the most important predictor for all stations in terms of TTPR is prefilling. For instance at station NSWE time lags of two and seven days lead to TTPRs of 85.71% and ($n_\epsilon = 2254$) and 84.12% ($n_\epsilon =$

2337), respectively. For all other stations TTPRs were also above 70% when using no timelag at all. It is also worth noting that the TTPR of prefilling shrinks consistently when increasing the time lag.

We can conclude that the choice of predictors depends on the station at hand. Depending on their location, values of predictors in the AoIs vary, especially when considering wind fields. For SP the model uses always low-pressure systems in order to achieve TPPs. Most valuable predictors are SP and U10, showing mainly low-pressure and west wind fields in AoIs. In general,

time lags up to three days were used. Choosing longer time lags often leads to worse results. Overall PF was the most useful predictor for all stations. The results are summarized in Fig. 11.

We did see that in some cases storm surges occur even though predictors show values one would not expect based on theory, e.g. high-pressure fields in the AoI. Physically it is not straightforward to explain. Note though that we used each predictor only in isolation. Hence, it might be possible that a combination of other predictors, e.g. a strong prefilling in combination with

west winds, is inducing the storm surge. Therefore, we will analyze combinations of predictors in the following experiments.

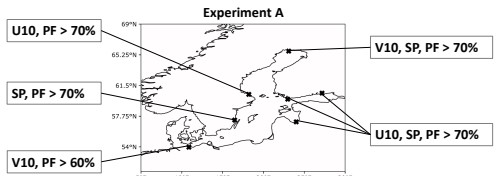

**Figure 11.** Summary of best predictors per station for experiment A. The percentage indicates the corresponding VTPR or TTPR.



## 5.2   B - Combination of all predictors

In this experiment, we combined all ERA5-predictors in order to rank them by feature importance and look at the behavior of their corresponding PMs (see Table A5). For almost all stations SP and U10 are the most important predictors. They again show pronounced low-pressure fields (below 980 hPa) and strong west winds (greater than $15\,\mathrm{ms}^{-1}$) in their respective AoI. Only for the stations at meridional extents of the Baltic Sea, this behavior switched and V10 becomes important as well. In terms of PMs the physical components did not change compared to experiment A.

Nevertheless using the predictors in combination showed an order of importance as depicted in Figure 12. We can see that SP and U10 are mostly used by the model, but it also switches depending on the station. In terms of the maximum VTPR achieved at each station, using isolated predictors lead to better results

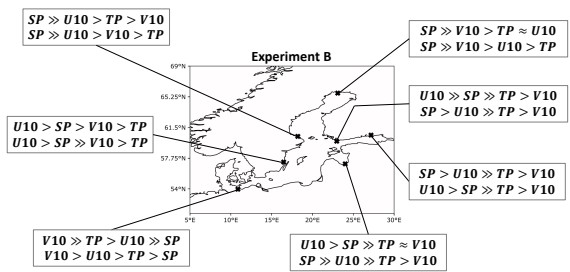

**Figure 12.** Order of predictor importance for experiment B with run_ids 1, 3. First and second row show $\hat{t} = 1, 2$, respectively. The $\gg$ sign indicates that the feature importance was almost double as high, the $\approx$ indicates approximate equality. For PF feature importance was close to zero.

## 5.3   C - Coupling of U10 and SP

We already observed that SP and U10 are important predictors. In theory resonance coupling of strong winds and moving weather systems (low-pressure systems) lead to extreme storm surges as well. Hence, we will now investigate several combinations of those predictors as shown in Table A6.

Combining both predictors with a similar time lag did not improve results compared to using them in isolation. For some stations (NSWE) it even leads to worse VTPRs, perhaps an indication of overfitting at those stations. Nevertheless, short-term combinations with time lags up to three days seem to work best. Only on a few occasions (station 1: FIN) time lags up to 5 days also produce acceptable results.

Best results could be observed for station 2 (FINBAY) with the highest VTPR of 75.23% using no time lag. For this station time lags of one or two days lead to VTPRs above 70%. All other stations had similar VTPRs above 60%, mainly for time lags up to three days. The only (expected) exception was station 4 (DEU) for which both predictors can not be used (TTPRs below 45%).





The PMs mainly showed similar behavior as for using isolated predictors. Again, an increased time lag leads to shifted AoIs. We expect the effects of U10 on storm surge to be slower than the influence of low-pressure systems. This is due to the fact that U10 needs to transfer kinetic energy to the Ocean's surface first in order to induce waves. Hence, we used a shorter time

lag for SP compared to U10 in the next set of the experiment.

Comparing VTPRs across all stations of both subsets of this experiment, we deduce that similar VTPRs over 60% and in best cases even up to 70% could be achieved. Using a difference in time lags of SP and U10 did lead to more stable results when altering the timelag of U10. In other words, the VTPR did not diminish quickly when increasing the time lag of U10.

In total this experiment showed, that for most stations a combination of short and long-term data as well as a positive difference

in time lags between U10 and SP leads to good results in terms of VTPRs (see Fig. 13).

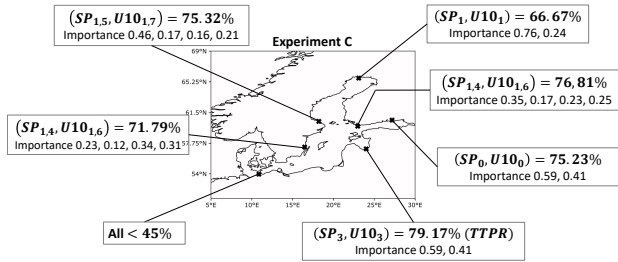

**Figure 13.** Best combinations of predictors for experiment C. Importances are ordered as subscripted time-lags. Depending on the station VTPR or TTPR is shown.





## 5.4   D - Combinations of Westwind-Timelags

As was already shown, west winds are an important driver of storm surges. If those winds blow consistently over several days, it deforms the sea surface and causes drift currents. Hence, in this experiment, we will investigate U10 with several time lag combinations as shown in Table A7.

We already know that a time lag of one or two days works well for U10 due to previous experiments. Hence we combined those short-term time lags with longer ones in the first subset of this experiment (run-ids 0 – 3). In a second subset, we investigated timelags up to a week, comparing short and long-term combinations of the time lag (run-ids 4 – 7). Finally, we spread the time lags over a whole week and even over a whole month for run-ids 8 – 11.

Overall using combinations of only U10 worked quite well for almost all stations, with VTPRs above 70%. Only for stations 0
(NSWE) and 5 (WSWE) the best VTPRs were just above 60%. As expected station 4 (DEU) was showing poor results. Mostly timelags up to four days worked the best for all stations.

AoIs and PMs show again similar behavior to experiments before, i.e. mainly strong west winds mostly in regions around the Danish Straits or Southern Baltic Coastline. Depending on the location of the station AoIs vary their area. They do so even for slight positional changes of stations, for instance stations 5 and 6. Figure 14 depicts this for a time lag of two days. While
for station 6 west winds around the North Sea entrance of the Danish Straits are important, this is not the case for station 5. The whole AoI shifts more towards the East. One explanation might be that west winds can not induce a direct wind-setup for station 6, as its coastline is oriented towards the North, hence sheltered from the winds. The opposite is true for station 5. It is completely directed towards the South and South-West. Hence, west winds may induce strong wind buildup here and prefilling induced by the wind around the Danish Straits is not as important for model predictions.

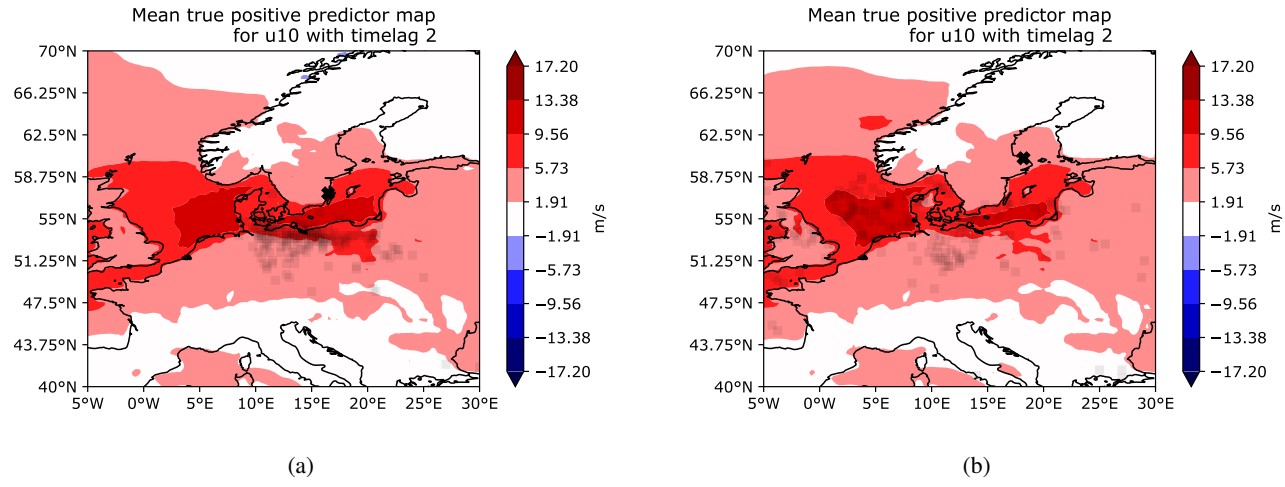

**Figure 14.** Mean Predictor Maps for TPPs using U10 with a time lag of 2 days at stations (a) WSWE and (b) WSWE2


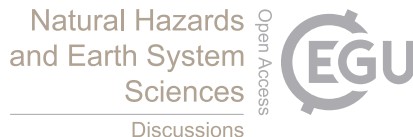

In summary combinations of U10 can be used for most stations as a good predictor when focusing on time lags up to four days (see Fig. 15). For some stations, time lags up to a week also lead to good predictions. Even longer timelags should not be used as they are mostly disregarded by the model.

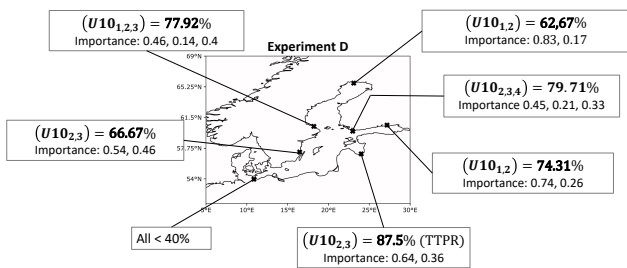

**Figure 15.** Best combinations of predictors for experiment D. Depending on the station VTPR or TTPR is shown. Importances are ordered as subscripted time-lags.


## 5.5 E - Predictor Combinations from Theory

We tried to emulate the effect of cumulative rain and looked into how information on prefilling changes the behavior of the
west wind for model predictions. The combinations of predictors can be found in Table A8 and results of VTPRs and TTPRs
are summarized in Fig. 16

Best results for the cumulative rain combination were observed for station 6 (WSWE2), with a VTPR of 74.03% ($n_\epsilon = 77$).
For stations FIN, FINBAY and WSWE also good results around 60% VTPR were calculated. When looking at the importance
though, one can see that mostly U10 is used for model predictions. For all stations except station NSWE the sum of TP feature
importances is smaller than the feature importance of U10.

The most interesting observations could be made when looking at station 1 (FIN) for combinations of U10 and PF. As the
theory suggests, with a state of prefilling in the Baltic Sea weaker west winds are needed to induce storm surges compared to
times without prefilling. The PMs of TPPs for an isolated U10 (as in experiment A), a combination of U10 and PF (run-id 1), as
well as their difference, are depicted in Fig. 17. In both cases, U10 was implemented with a time lag of three days. Comparing
the area around the Danish Straits, i.e. 2°E – 15°E and 54°N – 57°N, shows that in the case of no prefilling west winds are
blowing up to 5 ms$^{-1}$ stronger in specific areas. Hence, the model predicts (on average) more storm surges with weaker west
wind correctly, when passing information of the prefilling to it.

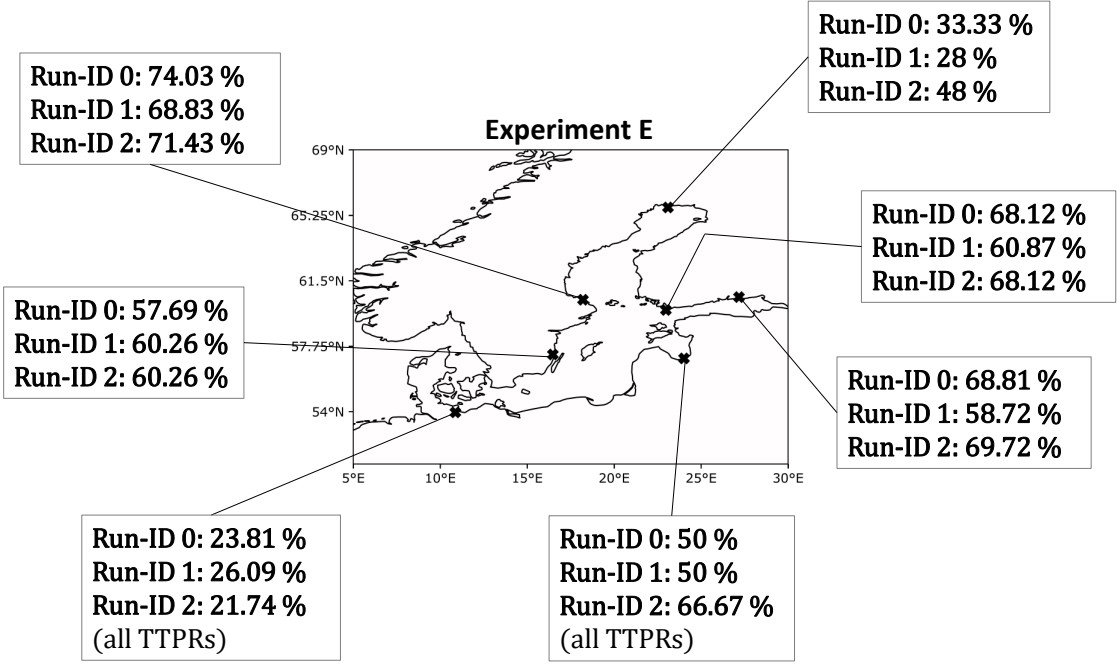

**Figure 16.** TTPR or VTPR results for all run-ids of experiment E.
**Figure 17.** Mean Predictor Maps of U10 at station 1 (FIN) and a time lag of 3; (a) Without information on prefilling, (b) with information on prefilling, (c) difference of (a) and (b).





## 5.6 F - Combinations of prefilling-timelags

Strongly influenced by strong west wind is the prefilling of the Baltic Sea. While Weisse (2014) and Mudersbach and Jensen
(2010) define the prefilling as the rolling mean of the water levels at Degerby over 20 consecutive days, we will use a time lag
of the records of water level at Degerby as the predictor. In this experiment, we investigate PF as an isolated predictor for time
lags up to a month as well as combinations of PF which include short-term (up to a week) as well as long-term (up to a month)
information on the water levels. All combinations can be found in Table A9.

When using isolated predictors, results show that shorter timelags work better in terms of TTPR than longer ones. For instance
TTPRs of station 2 (FINBAY) for time lags of 10, 15, 20, and 25 days were 71.12%, 63.63%, 49.85%, 61.61%, respectively.
Combining the information of several days leads to best results for all stations. The overall highest TTPR of 91.15% ($n_\epsilon =$
2869) could be achieved for station 1 (FIN) when using the time lag-combination of 3, 14, 21 and 30 days. In this case, the
feature importance for 3 days was significantly higher than the one for 14 days, which itself was more than the doubled feature
importance of $\hat{t} = 21$. This indicates that the model heavily relies on most recent water level recordings in order to provide
TPPs, a behavior that is repeated generally for all stations.

Altogether prefilling seems to be a good predictor for almost all stations when combining information from the previous water
records with records up to two weeks. Independent of the station, combining several time lags of information works better than
using the information in isolation.





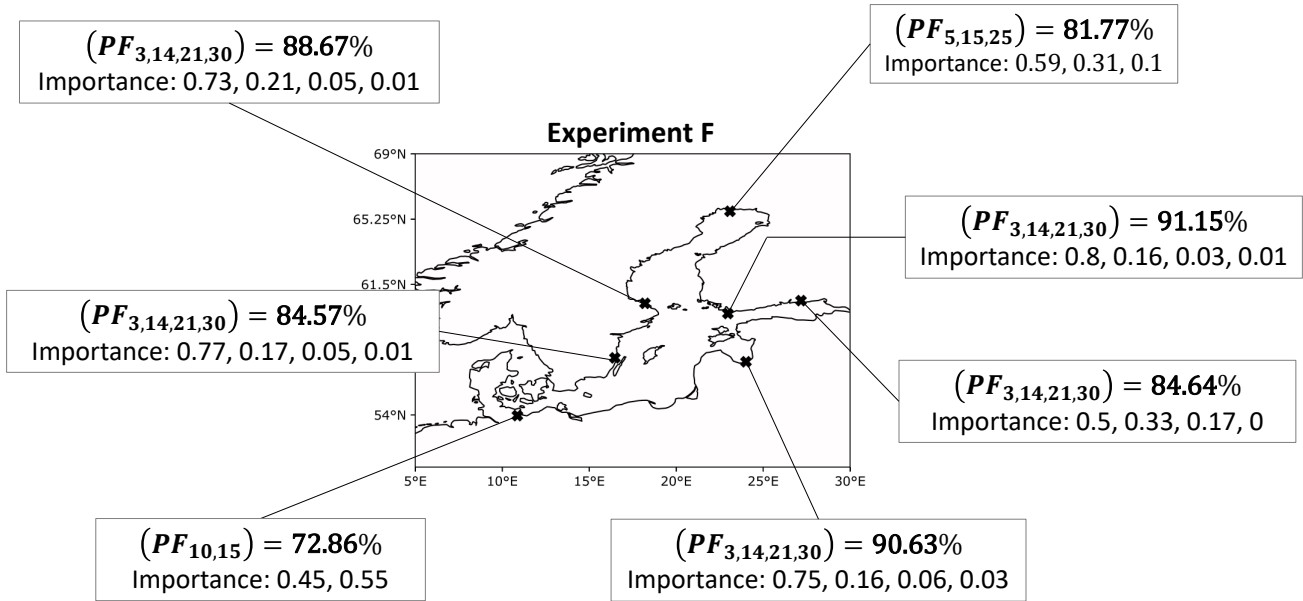

**Figure 18.** Best combinations in terms of TTPR of PF for experiment F. Importances are ordered as subscripted timelags.





## 6 Discussion

The theory indicates that one predictor alone should not be sufficient to describe storm surges. The main features are the wind stress and the low-pressure systems (below 980 hPa) as well as their speeds. While our models showed also good results when using predictors isolated, they showed more robust results when using them in combination. Furthermore, for almost all stations (except station 4) surface pressure and west wind were the most important ERA5-predictors. We could verify that mostly low-pressure fields below 980 hPa and strong (mean) west winds of 10 ms$^{-1}$ around the area of the Danish Straits lead

to TPPs, especially for stations located in the Northeast of the Baltic Sea. For those stations the AoI of U10 was situated South of the Danish Straits reaching inland towards mid-Germany. This can actually be explained by predominant South-Westerly winds in winter months, which eventually push water masses towards the Northeast. Furthermore PMs showed (when looking beyond the AoI) that those strong west winds often acted on a large horizontal distance, which according to Weisse and von Storch (2010) increases the potential of storm surges. It is this very wind direction that leads to the fact that U10 as well as

SP did not lead to any good predictions for station 4 (DEU). This is theoretically sound as for stations in the Southwest of the Baltic Sea water is pushed away towards the Northeast due to winds and baric waves. By contrast, those stations should be more subject to negative storm surges, which we did not investigate in this study.

For southern stations rather north-easterly wind should be a predominant factor. We saw this for station DEU, where the most important predictor in terms of feature importance was V10.

According to Leppäranta and Myrberg (2009) the largest amount of precipitation is found at the eastern coast of the Baltic Sea due to the winds blowing mostly eastward in wintertime. We could not recover this for our model. If any structure at all could be obtained from AoIs of TP it was the importance around the area of Bergen and the UK. Also corresponding PMs of TP did not show stronger rain in the eastern coast of the Baltic Sea. In contrast Gönnert et al. (2001) states that the influence of precipitation is not directly related to storm surge magnitudes, but rather alters preconditions like the prefilling of the Baltic

Sea and the filling of rivers and estuaries. Together with the fact that west winds are not as strong when a condition of prefilling exists, prefilling itself should be of great usage as a predictor. For almost all stations this was actually true. Compared to other ERA5-predictors PF was generally leading to better VTPRs.

Sometimes our model showed patterns for AoI and PMs though, that was hard to explain by theory. For instance, for station NSWE low pressure fields in the European North Sea were of great importance, instead of low-pressure systems close to the

station. This behavior showed mainly when using time lags of several days. Theoretically low pressure systems in those areas move towards the East, i.e. in the direction of the station, which might be one possible explanation. Additionally, for some cases (FNPs) there were storm surges even though high-pressure fields were present. We first argued that this might be due to the lack of combining predictors in model runs but the same behavior reappeared when using combinations of predictors. At this point, we do not have any valid explanation for this behavior. One idea is that the involved time lag of predictors was long

enough, such that high-pressure systems could turn into low-pressure systems before the actual day of the storm surge. One could account for this when calculating hourly gradients of atmospheric pressure, which indicate a rapid (de-) intensification of low-pressure systems (similar to Bruneau et al. (2020)).





Nevertheless, we saw that time lagging the predictors increased model results. This is in alignment with Tyralis et al. (2019), who showed that Random Forests worked better when time-lagged predictors were used. In general timelags up to 4 days worked quite reasonably, while longer timelags did not add much value to VTPRs. For instance, a time lag of 2 or 3 days for U10 was often the best choice. This is what we expected, especially for north-eastern stations as deep-water waves need approximately 2 days to travel across the Baltic Sea Furthermore, we saw that implementing a longer time lag for U10 compared to SP leads to good results in terms of VTPRs when using both in combination. For PF mostly short-term time lags work best but still it was possible to even increase the timelag up to a week. This is contradicting the actual definition of prefilling and one might argue against the usage of the plain timeseries of water recordings at Degerby as a plausible predictor.

Some caveats of our model need to be mentioned. First of all we only use a period of 3 months over 9 years to generate train and test data. But Bruneau et al. (2020) showed that for Machine Learning, specifically Artificial Neural Networks, 6-7 years of daily training data is necessary. We only used a total of 18 months though. In order to overcome this, one could extend the dataset to longer time periods. Using more data increases computing time though, which is one reason why we did not implement it. Furthermore, models trained with predictors based on remotely sensed data outperformed models forced with predictors obtained from reanalysis data (Tyralis et al., 2019). We used only reanalysis data as predictors. If data sources with remotely sensed data are available it would be better to test the model on them.

For future studies within this context, it would be interesting to alter and specify some of the predictors. For instance, instead of only using U10 and V10 one could actually calculate all of the wind stresses, i.e. the wind direction, wind velocity and its duration. Our dataset did not involve the duration, which is especially important for the generation of surface waves and swell. Furthermore, we did not use wind directions per se as a predictor but rather the zonal and meridional wind speeds. One could calculate wind directions of those datasets and use it as a new predictor. Similarly, if low-pressure systems move at relatively high velocities, i.e. greater than 16 ms$^{-1}$, a sub pressure-driven storm surge occurs (Wolski and Wisniewski, 2021), because the effect of the baric wave is stronger than the one of the wind. We did not use the speed nor the trajectory of a low-pressure system as model input. But this can be important as it induces resonance coupling and gives direction to the induced baric wave. Closely connected to these topics are climate modes like the NAO. In theory, the NAO-index is correlated to prefilling and the strength of westerlies. It would be interesting to use the NAO-Index or BANOS-Index as a predictor as well. Another physical change that can be made is to look at negative storm surges instead of positive ones and see if behaviors of U10 and SP change for stations like DEU. For instance, the bays of Mecklenburg and Kiel ran into strong negative storm surges due to water outflow caused by low-pressure systems moving towards the East (Wolski and Wisniewski, 2020).

From a technical perspective one could adjust the definition, i.e. the one hot encoding, to represent the alarming levels of specific stations instead of using percentiles. It would also be highly interesting to extend the usage of the RF to Random Regression Forests in order to investigate and predict actual heights of water level during storm surges. Further Tiggeloven et al. (2021) showed promising results using Deep-Learning-Methods when those models are tailored for specific regions. Hence, complicating the model architecture can also lead to promising results.



## 7 Conclusion

In this study, we designed a prediction scheme for the occurrence of storm surges, i.e. the top daily 5% coastal water levels, for 7 stations across the Baltic Sea. The prediction horizon is a few days and the method is based on a Random Forest used as a binary classifier. The method was tested on records of the water level at respective stations from GESLA3 and atmospheric

predictors were taken from the ERA5 dataset, from which we choose variables of surface pressure (SP), zonal (U10) and meridional (V10) windspeeds at 10 meter above the Earth's surface and total precipitation (TP). Despite its relative simplicity, the purely data-driven Random Forest binary classifier is able to predict the occurrence of storm surges in the Baltic Sea with a few days lead time with high sensitivity. The method is able to identify the relevant predictors and the relevant regions among a set of atmospheric variables, agreeing with physical expectations. The RF method is able to discriminate the predictors according

to the station location. For stations at zonal extends of the Baltic Sea, U10 and SP were the most important predictors, showing strong west winds and pronounced low-pressure systems when modelling extreme storm surges. For stations at meridional extends the importance of V10 increases.

Westwind around the Danish Straits often indicated the onset of an extreme storm surge, probably due to its influence on the Baltic Seas prefilling. We could also recover the fact that with increased prefilling the importance of west winds tend to be

weaker in cases of storm surges. Increasing the lead time of predictors decreased model accuracy. Mostly, the method works well for lead times of up to three days. Combining several time lags of information works better than using the different lead time information in isolation.

With respect to the very short computing time of these models (up to 10 minutes per station) and their high sensitivity, further investigations of machine learning methods in the context of climate-extreme event predictions are recommended.


*Code and data availability.* The code is available at the reference Bellinghause (2022). The ERA5 data are publicly available from reference Hersbach et al. (2018). The GESLA data are available at https://gesla787883612.wordpress.com



**Appendix A: Tables**

| Number of station | GESLA code | Identifier |
|:---:|:---:|:---:|
| 0 | "kalixstoron-kal-swe-cmems" | NSWE |
| 1 | "hanko-han-fin-cmems" | FIN |
| 2 | "hamina-ham-fin-cmems" | FINBAY |
| 3 | "daugavgriva-dau-lva-cmems" | LVA |
| 4 | "travemuende-tra-deu-cmems" | DEU |
| 5 | "oskarshamn-osk-swe-cmems" | WSWE |
| 6 | "forsmark-for-swe-cmems" | WSWE2 |

**Table A1.** Number of stations as in Fig 2 and corresponding code in GESLA dataset.




| Name | Units | Short Description |
|------|-------|-------------------|
| sp | Pa | Pressure (force per unit area) of the atmosphere on the surface of land, sea and in-land water. It is measured by the weight of total air in a vertical column above the area of the Earth's surface. |
| tp | m | Accumulated liquid and frozen water that falls to the Earth's surface. It represents the sum of large-scale precipitation and convective precipitation. The units indicate the depth the water would have when evenly spread over the grid box. |
| u10 | $\mathrm{ms}^{-1}$ | Eastward component of the 10m wind, i.e. the horizontal speed of air moving towards the east at a height of ten metres above the Earth's surface. |
| v10 | $\mathrm{ms}^{-1}$ | Northward component of the 10m wind, i.e. the horizontal speed of air moving towards the north at a height of ten metres above the Earth's surface |

**Table A2.** Variables of ERA5 dataset used as predictors. Description of data is taken from the parameter database of the official ECMWF website.





| Parameter | Value | Short Description |
|-----------|-------|-------------------|
| n_estimator | [333, 666, 1000] | Number of DTs used within a RF. |
| max_depth | [1, 2, 3] | Depth of each DT. |
| class_weight | "balanced" | Associated weighting of each class. |
| oob_score | "True" | Calculating out-of-bag sample scores for each DT. |
| optimizer | "RandomSearchCV" | Functionality to find best combination of hyperparameters. Optionally "GridSearchCV" can be used. |
| k | 3 | k-fold cross-validation used by optimizer. |
| n_iter | 100 | Number of parameter settings that are sampled by "RandomSearchCV". Trades off runtime against quality of the solution. |

**Table A3.** Parameters used to find optimal hyperparameters of the random forest. When multiple values are given, the optimizer chooses the best combination amongst those.





| Experiment: **A** | | |
|---|---|---|
| **Run_Id** | **Predictors** | **Timelags (in days)** |
| 0 – 4 | SP, TP, U10, V10, PF | no timelag, i.e. 0 |
| 5 – 9 | SP, TP, U10, V10, PF | all with timelag 1 |
| 10 – 14 | SP, TP, U10, V10, PF | all with timelag 2 |
| 15 – 19 | SP, TP, U10, V10, PF | all with timelag 3 |
| 20 – 24 | SP, TP, U10, V10, PF | all with timelag 4 |
| 25 – 29 | SP, TP, U10, V10, PF | all with timelag 5 |
| 30 – 34 | SP, TP, U10, V10, PF | all with timelag 6 |
| 35 – 39 | SP, TP, U10, V10, PF | all with timelag 7 |

**Table A4.** Parameters and timelags used for experiment **A**. All predictors are used in isolation, no combinations are used.





| Experiment: **B** | | |
|:---:|:---:|:---:|
| **Run_Id** | **Predictors** | **Timelags (in days)** |
| 0 | (SP, TP, U10, V10) | (1, 1, 1, 1) |
| 1 | (SP, TP, U10, V10, PF) | (1, 1, 1, 1, 1) |
| 2 | (SP, TP, U10, V10) | (2, 2, 2, 2) |
| 3 | (SP, TP, U10, V10, PF) | (2, 2, 2, 2, 2) |

**Table A5.** Parameters and timelags used for experiment **B**. Parenthesis indicate that predictors are used in combination.





| Experiment: **C** | | |
|---|---|---|
| **Run_Id** | **Predictors** | **Timelags (in days)** |
| 0, 1, … , 7 | (SP, U10), (SP, U10), … , (SP, U10) | (0, 0), (1, 1), … , (7, 7) |
| 8, 9, 10 | (SP, U10), (SP, U10), (SP, U10) | (2, 3), (2, 4), (2, 5) |
| 11 | (SP, SP, U10, U10) | (1, 3, 1, 5) |
| 12 | (SP, SP, U10, U10) | (1, 4, 1, 6) |
| 12 | (SP, SP, U10, U10) | (1, 5, 1, 7) |

**Table A6.** Parameters and timelags used for experiment **C**. Parenthesis indicate that predictors are used in combination.



| Experiment: **D** | | |
|---|---|---|
| **Run_Id** | **Predictors** | **Timelags (in days)** |
| 0 – 3 | all (U10, U10) | (1, 2), (2, 3), (2, 4), (3, 6) |
| 4 – 7 | all (U10, U10, U10) | (1, 2, 3), (2, 3, 4), (3, 4, 5), (5, 6, 7) |
| 8 – 11 | all (U10, U10, U10, U10) | (1, 2, 3, 4), (4, 5, 6, 7), (1, 3, 5, 7), (1, 7, 14, 21) |

**Table A7.** Parameters and timelags used for experiment **D**. Parenthesis indicate that predictors are used in combination.





| Experiment: **E** | | |
| --- | --- | --- |
| **Run_Id** | **Predictors** | **Timelags (in days)** |
| 0 | (TP, TP, TP, U10) | (7, 5, 2, 2) |
| 1 | (U10, PF, PF, PF) | (3, 7, 5, 2) |
| 2 | (U10, U10, PF) | (5, 2, 7) |

**Table A8.** Parameters and timelags used for experiment
**E**. Parenthesis indicate that predictors are used in combi-
nation.



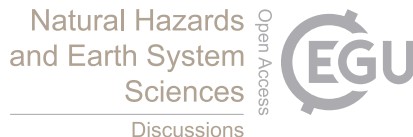

| Experiment: **F** | | |
|---|---|---|
| **Run_Id** | **Predictors** | **Timelags (in days)** |
| 0 – 3 | all PF | 10, 15, 20, 25 |
| 4 – 6 | all (PF, PF) | (5, 10), (10, 15), (20, 25) |
| 7, 8 | all (PF, PF, PF) | (5, 15, 25), (7, 14, 21) |
| 9 | (PF, PF, PF, PF) | (3, 14, 21, 30) |

**Table A9.** Parameters and timelags used for experiment **F**.
Parenthesis indicate that predictors are used in combination.





*Author contributions.* All authors contributed to develop the original research goal, analysed and discussed the results. K.B. coded the software, carried out the data analysis, and drafted the initial versions of the manuscript. B.H and E.Z contributed to the later and final version of the manuscript.

*Competing interests.* We declare that we have no competing interests.

*Acknowledgements.* The ERA5 data (Hersbach et al., 2018) were downloaded from the Copernicus Climate Change Service (C3S) Climate Data Store. The results contain modified Copernicus Climate Change Service information 2020. Neither the European Commission nor ECMWF is responsible for any use that may be made of the Copernicus information or data it contains.

We also thank the GESLA project ( https://gesla787883612.wordpress.com/ ) for making the extreme sea level data sets available for the
scientific community.



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
