# Peer review of "Short-term prediction of extreme sea-level at the Baltic Sea coast by Random Forests"

_Natural Hazards and Earth System Sciences, 2023_

## Referee Comment (RC1)

Review report on NHESS-2023-21
Short-term prediction of extreme sea-level at the Baltic Sea coast by Random Forests
by Bellinghausen, K. et al.

The authors investigate (deterministic) random forest classifier models for predicting "extreme" sea level at the Baltic sea (defined as exceedances of the 95th percentile). Various meteorological data from the ERA5 dataset serves as input to predict the sea-level extremes at seven measurement stations. While the topic is generally interesting and within the scope of the journal, I have several major concerns regarding the paper, including the choice of reference methods, the overall presentation of the results, and the lack of details regarding the description of the model setup.

Major comments

1. The model descriptions and setup lacks some crucial details, includes partially incorrect statements and seems partially somewhat questionable:
   a. The description of the input predictors in line 223f is not sufficiently clear. What is actually being used as input predictors for the meteorological variables? Those are 2D fields, but are averages computed over the whole field (e.g. over all grid points and hourly data)? Would it not have been an alternative to use the hourly values of only those hours corresponding to the relevant occurrence of the storm surge (the maximum over the day, as defined before)? Or alternatively, some other summary statistics of the forecast fields, such as either local values close to the station of interest, or averages over different sub-domains?
   b. If all missing values are set to -999, doesn't this imply problematic behavior since they would potentially be grouped together with other "low" values of the corresponding predictor variable (unless the RF specifically accounts for missingness, which seems unlikely given the limited complexity of the trees chosen in the study)?
   c. line 270: Trees up to a maximum depth of only 3 are considered. In my view, this leads to trees that are less complex than usually applied in many practical applications. The complexity of individual DTs is usually controlled via the minimum node size, which tends to be something small, typically below 50. What typical minimum node size do you see for the chosen maximum depth, and how do results for more complex DTs compare?

2. In light of the previous comment, since I assume the averaging is done over all grid points and hours for the input predictors, I find the figures showing full meteorological fields, which are then used to infer the meteorological interpretation of the importance of this predictor somewhat misleading. While often tendencies and values in specific regions are discussed, the model only receives the overall average as an input.

3. A main limitation of the study is that only simple binary predictions of storm surges are considered. Over the past two decades, large parts of the meteorological and forecasting literature have transitioned towards probabilistic predictions, and random forest enable (for example) probabilistic classifier models in a straightforward manner.

4. The study does not consider any competitive benchmark models – neither a simple non-ML classification model (e.g. logistic regression), nor a climatological forecast or an operational storm surge model. At the very least, a simple climatological model that accounts for the seasonality in the target variable would have been essential to consider as a naive

benchmark to allow for a fair assessment of the RF models. For now, the models are compared to benchmarks that either always predict a 1 or a 0 only.

5. Section 5 contains the main results of the paper, which are organized via "case studies" of comparisons of individual model configurations based on different combinations of input variables. As noted elsewhere, in principle I find this approach of connecting model performance and physical "interpretation" interesting, however, this makes the results section cumbersome to read and difficult to follow, since it is challenging to keep track of all the different comparisons and model variants (for example since a quantitative overall comparison is missing). A more common approach in the ML literature would have been to simply supply all the considered input predictors (the number of which would still be manageable) to the RF model, and learn the model itself which combinations and connections are important. Then, in a second step, feature importance methods can be used to analyze what the model has learned and this can be connected to the meteorological domain expertise.

Minor comments

1. line 35: Wouldn't it have been an interesting alternative to compare directly to an operational storm surge forecasting system?

2. line 55: "... but rather try to identify recurring patterns in the data…": Isn't that the same as what the statistical models do?

3. The description of the meteorological background in the introduction is fairly long. Overall, the authors seem to have had the intention of combining a detailed meteorological analysis of the physical processes relevant to the prediction problem at hand, with a detailed study of random forest based forecasting methods. While this certainly is an interesting idea, it makes the paper fairly lengthy and not easy to follow, see also the comments on the presentation of the results.

4. It does not become clear from Section 2.2 what the predictand actually is, what kind of predictions are sought for (deterministic / probabilistic etc).

5. In my view, the setup is sufficiently simple such that Figure 3 could be left out without losing any relevant information.

6. Personally, I don't see the particular relevance of using this exact version of Figure 4. It does not directly fit the setting here ("averaging" is not really appropriate in a binary prediction setting). Further, it remains unclear whether the image, or the architecture was taken from the linked website. I would have preferred a merged version of Figures 4 and 5, which is more adapted to the situation at hand.

7. The descriptions of random forests needs improvements, as it lacks details and is partially incorrect:
   a. line 249: Typically, all the round nodes in the lowest row in Figure 5 would be called terminal nodes / leafs.
   b. line 252: "... random sample of the *test* data". This should be "training data" instead.
   c. line 254: It should probably also be explained how a single DT arrives at a binary decision, not only the final RF.

d. line 257: Does this mean you re-train the models to compute the importances? A computationally less costly and often applied alternative is to use permutation-based methods that do not require re-training.

e. line 269f: "... will lead to less overfitting": In my view, this is only true for very low numbers of trees. Otherwise, with a growing number of trees, the RF model will converge to the "true tree model", but this will not have a clear effect on overfitting. A reference should be provided here if you disagree.

f. line 278: Why is the oob_score parameter important here? This (to me) seems to be relevant only when the number of trees is based monitoring the oob estimates of the forecast error, which is not done here.

8. line 341: "... contains more instances … due to the hourly recording time": This is not clear to me, wasn't the target aggregated to daily values? Why do the time scales not match here?

9. The discussion of the results in Section 5 often talks about good / bad models, without giving a clear indication of what specific true positive / false negative / … rates are required for a model to be "good". Again, this indicates that a sensible, non-naive benchmark is missing.

10. The discussion/conclusion is missing some summary of the overall findings: If I were to develop a RF model for storm surge extremes, what predictors and model setup should I use?

Technical comments

1. line 7f: The discussion of the cross-validation setup does not need to be part of the abstract

2. line 34-35: What is the difference between forecasting and predicting, and why is it relevant here?

3. line 53/54: Abbreviation "ML" introduced twice.

4. line 62: Reference should be in parentheses.

5. Figure 1: Reference should not be in parentheses.

6. line 260: There is a "?" after "small"

7. Figure 8: What does that t hat refer to?

8. line 528: increased -> improved